# PtychoFormer: A Transformer-based Model for Ptychographic Phase Retrieval

## Abstract

Ptychography is a computational method of microscopy that recovers high-resolution transmission images of samples from a series of diffraction patterns. While conventional phase retrieval algorithms can iteratively recover the images, they require oversampled diffraction patterns, incur significant computational costs, and struggle to recover the absolute phase of the sample's transmission function. Deep learning algorithms for ptychography are a promising approach to resolving the limitations of iterative algorithms. We present PtychoFormer, a hierarchical transformer-based model for data-driven single-shot ptychographic phase retrieval. PtychoFormer processes subsets of diffraction patterns, generating local inferences that are seamlessly stitched together to produce a high-quality reconstruction. Our model exhibits tolerance to sparsely scanned diffraction patterns and achieves up to 3600 times faster imaging speed than the extended ptychographic iterative engine (ePIE). We also propose the extended-PtychoFormer (ePF), a hybrid approach that combines the benefits of PtychoFormer with the ePIE. ePF minimizes global phase shifts and significantly enhances reconstruction quality, achieving state-of-the-art phase retrieval in ptychography.

## 1 Introduction

Microscopic and sub-microscopic imaging of tissue, cells, proteins, and crystals are a crucial tool in biological and materials sciences. Optical microscopic imaging relies on expensive lenses and capturing devices and is limited by the diffraction limit and lens aberrations. Furthermore, optical imaging requires high-contrast material, necessitating staining of transparent materials, making it impractical for live cell imaging. Coherent diffractive imaging (CDI) is a "lensless" approach to imaging small, transparent objects by capturing downstream diffraction patterns free of lens aberrations. Ptychography is a natural extension of CDI that captures high-resolution images by illuminating a sample at multiple scan points with a coherent light probe. The coherent source for ptychography can be visible light, x-rays, or even electrons. Electron ptychography has captured some of the highest-resolution atomic images ever recorded (Jiang et al., 2018).

During ptychography, the sample is illuminated by the coherent probe at multiple scan points, and diffraction patterns of overlapping regions are captured. Since only the intensity of the interference can be captured downstream, losing the phase measurement, retrieving the sample's image poses a classic inverse problem. Phase retrieval algorithms, like the ptychographic iterative engine (PIE) (Rodenburg & Faulkner, 2004) and its variants, use the diffraction patterns to iteratively recover the sample's image as a transmission function. This transmission function comprises of an amplitude, representing the intensity distribution of the sample, and a phase, which carries information about the sample's refractive index variations and internal material properties.

Conventional phase retrieval algorithms can recover the transmission function but have significant drawbacks. Firstly, their slow, iterative computation, typically taking minutes to hours to converge, makes real-time imaging impractical. Secondly, these algorithms require heavily dense scanning, which causes numerous data collection to image a sample. Thirdly, the recovered transmission function is prone to global phase shift, resulting in inaccurate phase recovery, as shown in the recovered phase using extended-PIE (ePIE) (Maiden & Rodenburg, 2009) in Figure 1. In contrast, deep learning (DL) based phase retrieval algorithms offer a significant speed advantage. These networks are commonly convolutional neural networks (CNNs) trained to recover the transmission function

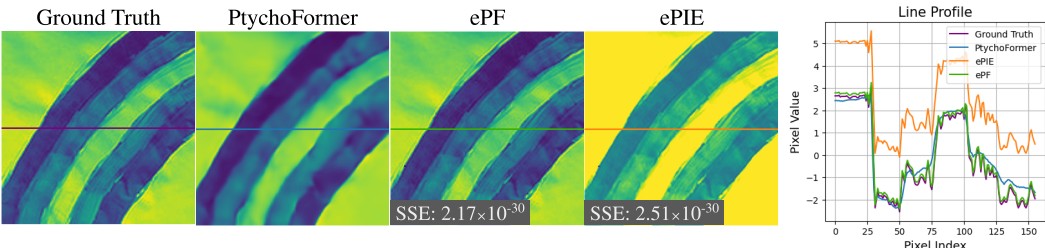

Figure 1: Comparison of the ground truth against the phase reconstructions from our proposed methods, PtychoFormer and extended-PtychoFormer (ePF), and ePIE. Sum squared error (SSE), the objective function for ePIE, cannot distinguish between globally shifted phase values, showing comparable SSE values between ePF and ePIE despite distinct line profiles. ePIE reconstruction is affected by a substantial global phase shift, while ePF achieves better estimation by leveraging PtychoFormer for initialization.

from the diffraction patterns in a single forward pass, enabling rapid reconstruction in under a second. DL methods also exhibit more tolerance to sparse scan patterns, reducing the need to capture numerous diffraction patterns for the region of interest (Guan & Tsai, 2019; Cherukara et al., 2020; Pan et al., 2023).

Despite these advancements, many existing DL methods, particularly those relying on CNNs, fail to account for the spatial relationships between overlapping diffraction patterns—a fundamental aspect of ptychography, noted by Konijnenberg (2017). This oversight limits the effectiveness of current models in capturing the intricate dependencies across diffraction patterns. Transformers, which have revolutionized natural language processing (NLP) and computer vision (CV) tasks by leveraging self-attention mechanisms (Jia et al., 2022), offer a promising solution. Their ability to capture long-range dependencies and contextual relationships (Vaswani et al., 2017) makes them well-suited for ptychography, where understanding the spatial dependencies of diffraction patterns is vital.

In our work **(a)** we introduce **PtychoFormer**, a hierarchical transformer-based architecture designed to scale effectively with large datasets, **(b)** Our input scheme **preserves the relative scanned position** of each diffraction pattern, and the architecture enables **spatial awareness** by leveraging this spatial information. **(c)** PtychoFormer outperforms previous DL phase retrieval methods in multiple phase and amplitude reconstruction tasks and is **up to 3600 times** faster than ePIE in our simulation. Finally, **(d)** we propose a hybrid approach, **extended-PtychoFormer (ePF)**, combining PtychoFormer with ePIE. ePF reduces NRMSE by **73.59%** for amplitude and **47.30%** for phase compared to ePIE while also minimizing the global phase shifts and accelerating convergence.

In **Section 2**, we describe the process of ptychography, the phase problem, and the ambiguity of the global phase shift. **Section 3** highlights related works in iterative and DL phase retrieval algorithms. **Section 4** details the formulation of PtychoFormer and ePF, and **Section 5** provides the empirical results by comparing them with other methods. Lastly, we discuss our findings and future directions in **Section 6** and **Section 7**, respectively.

## 2 BACKGROUND

In ptychography, the imaging sample is described by the complex-valued transmission function, $T(x, y)$. At any spatial coordinate $(x, y)$, the transmission function is defined in terms of its amplitude, $A(x, y)$, and phase, $\phi(x, y)$, as

$$T(x, y) = A(x, y) \cdot e^{i\phi(x, y)}. \tag{1}$$

The strict oversampling condition, where adjacent illumination areas must significantly overlap when capturing the diffraction patterns, is essential to recovering $T(x, y)$. This stems from a well-known constraint of light detectors, like the charged-coupled devices (CCDs) or photographic plates, which only measure the light intensity of the exiting light and lose the phase (i.e. the phase problem).

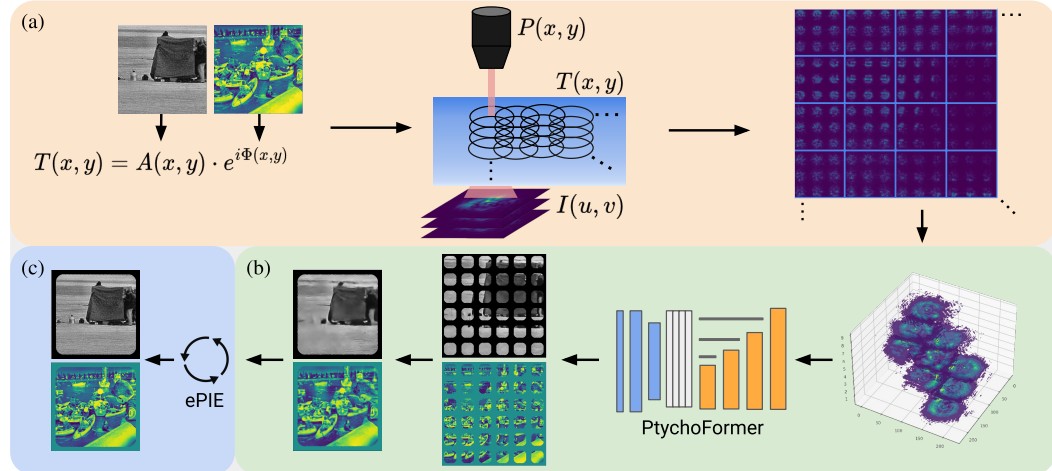

Figure 2: Comprehensive overview of the simulation using PtychoFormer and extended-PtychoFormer (ePF). (a) depicts the transmission function $T(x, y)$ characterized by its amplitude $A(x, y)$ and phase $\phi(x, y)$. The light probe $P(x, y)$ propagates through the sample to produce the diffraction pattern $I(u, v)$ in the far field. $I(u, v)$ are then grouped into sets of nine and placed in separate channels. (b) illustrates how PtychoFormer processes the sets in parallel to reconstruct local patches of $T(x, y)$ and then stitches them to complete the reconstruction. ePF framework builds on this approach by introducing an additional step (c), where the initial estimate from PtychoFormer is fed into ePIE for iterative refinement, further improving the reconstruction accuracy of $T(x, y)$.

The measured intensity of the diffraction pattern, $I_j(u, v)$, is defined as

$$I_j(u, v) = \left| \mathcal{F} \left\{ T(x, y) \cdot P(x - X_j, y - Y_j) \right\} \right|^2 , \tag{2}$$

where $\mathcal{F}$ denotes the Fourier transform, $P(x, y)$ is the complex probe function, $(u, v)$ is the reciprocal space coordinates, and $(X_j, Y_j)$ are the lateral offsets at the $j$th scan position. Phase retrieval algorithms resolve the phase problem by numerically approximating the lost phase using the oversampled patterns. Without oversampling, the reconstruction would suffer from the ambiguities of the phase problem (Shechtman et al., 2015).

The global phase shift is a trivial ambiguity in the reconstructed transmission function, $\hat{T}(x, y)$, that has yet to be solved. It accounts for additional constant and linear terms to the phase, as

$$\hat{T}(x, y) = A(x, y) \cdot e^{i\phi(x,y)} \cdot e^{i(a+bx+cy)} = A(x, y) \cdot e^{i(\phi(x,y)+a+bx+cy)}, \tag{3}$$

where $a$, $b$, and $c$ are real-valued coefficients. The constant phase term, $a$, emerges from the phase problem, where the magnitude of the sample's phase is uncertain during reconstruction, allowing $a$ to take any value in $\hat{T}(x, y)$. The linear phase terms, $b$ and $c$, can appear when the diffraction patterns are off-centered in the computational window in Fourier space (Guizar-Sicairos et al., 2011). Hence, transmission estimates from conventional phase retrieval algorithms do not contain the correct absolute phase values and instead provide only relative phase shifts, due to the ambiguity introduced by the global phase shift.

## 3 RELATED WORK

In this section, we discuss the current state of conventional phase retrieval algorithms and existing DL phase retrieval algorithms for ptychography.

**Iterative Phase Retrieval Algorithms.** The PIE algorithm (Rodenburg & Faulkner, 2004) laid the groundwork for phase retrieval algorithms in ptychography. Maiden & Rodenburg (2009) further advanced this approach with the ePIE algorithm, which improved convergence and reconstruction quality by iteratively approximating the transmission and probe function from the diffraction patterns. While several adaptations have since been developed, like mPIE (Maiden et al., 2017), MAIC-

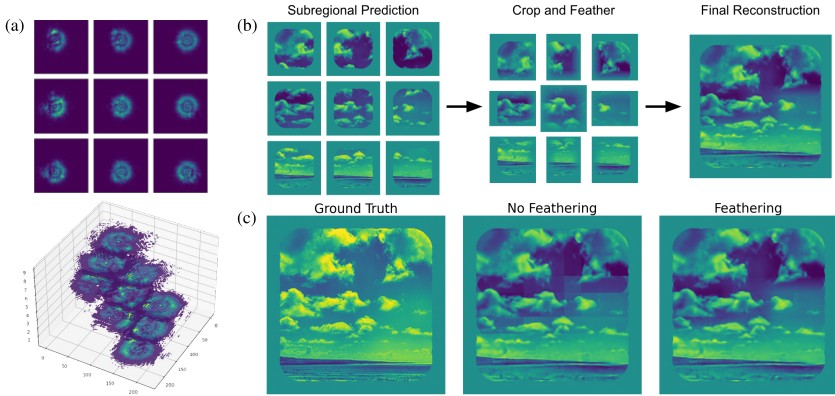

Figure 3: Input scheme depicted in (a) groups the diffraction patterns into subsets of nine and are placed in separate channels. This way, the spatial relation between each pattern is preserved. (b) showcases the stitching process, where Local predictions are cropped and feathered at the edges. (c) compares the reconstructions with and without feathering, whereby feathering effectively eliminates the grid artifacts present in the reconstruction without feathering.

PIE (Dou et al., 2020), and refPIE (Wittwer et al., 2022b), these algorithms are slow while requiring precise parameter tuning and densely scanned diffraction patterns for optimal performance.

**Deep Learning Phase Retrieval Algorithms.** DL phase retrieval algorithms eliminated the need for parameter tuning and dense scans, while substantially improving the reconstruction speed—enabling real-time imaging (Babu et al., 2023). However, many CNNs proposed for phase retrieval, such as PtychoNet (Guan & Tsai, 2019), PtychoNN (Cherukara et al., 2020), and PtyNet (Pan et al., 2023), process diffraction patterns one at a time, preventing them from leveraging the indispensable information from neighboring patterns. Gan et al. (2024) proposed PtychoDV to address this limitation by using a ViT and a deep unrolling network to process multiple diffraction patterns, each as a one-dimensional latent vector, with its scanned coordinates as positional embedding. Although injecting coordinates provides a rough spatial indication, coordinates alone inadequately capture the overlap between the vectorized patterns with high granularity.

## 4 PTYCHOFORMER

We detail the pre and post-processing, and architectural details of PtychoFormer and extended-PtychoFormer in this section. To reconstruct the phase and amplitude of a sample, multiple diffraction images are pre-processed by arranging them according to relative spatial positions before inputting to our transformer-based model. Local sub-regional predictions from the model are then seamlessly stitched together via feathering to minimize grid artifacts. Optionally, the predicted outputs are iteratively refined using ePIE in the extended-PtychoFormer, our hybrid approach. Figure 2 provides an overview of our reconstruction process.

**Input Processing.** Our devised input scheme, depicted in Figure 3(a), handles up to nine spatially overlapping diffraction patterns, each separated into distinct channels. The diffraction patterns are arranged according to their relative scanned coordinates, maintaining the inherent overlap between adjacent patterns. This organization allows the model to focus on learning the informational and positional dependencies between the patterns. This input scheme has been tested on various scan patterns and with different numbers of diffraction patterns, shown in Figure 5(b).

**Model Architecture.** The PtychoFormer architecture, illustrated in Figure 4 and detailed in Appendix A, leverages a Mix Transformer (MiT) encoder from the SegFormer architecture (Xie et al., 2021), along with a convolutional decoder to achieve efficient phase retrieval. In our research, we encountered several challenges that shaped our architectural choices.

The primary challenge was devising an encoder that effectively captures long-range dependencies between adjacent diffraction patterns. While CNNs excel at local feature extraction, they have

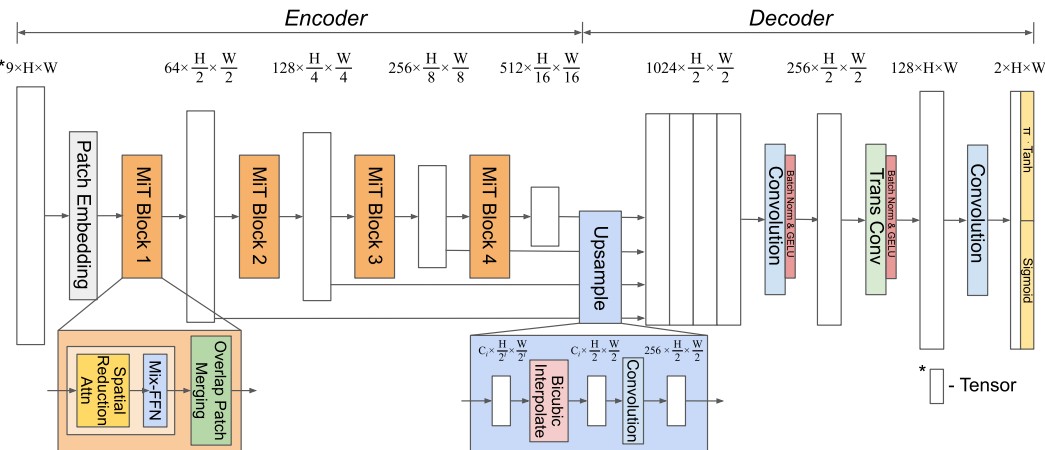

Figure 4: PtychoFormer leverages a Mix Transformer (MiT) encoder and a convolutional decoder. The encoder includes four MiT stages that progressively reduce spatial resolution while increasing feature channels. The decoder upsamples the encoder outputs, adjusts feature channels, and refines the resolution to produce the amplitude and phase estimates.

limited receptive fields without substantially increasing the convolutional window size. In contrast, transformers, with their self-attention mechanisms, naturally capture long-range dependencies, which is crucial for understanding the relationships between diffraction patterns in our research.

Employing standard transformers like the Vision Transformer (ViT) (Dosovitskiy et al., 2021) introduces new challenges. (1) ViT extracts only a single-resolution feature map, lacking the expressiveness to capture informative representation of the diffraction pattern to generate the transmission function. (2) Self-attention has quadratic time and space complexity relative to input length, creating a computational bottleneck that slows inference speed. (3) ViT relies on fixed positional encoding (PE) to retain spatial information of input patches, inhibiting innate handling of multi-resolution inputs and degrading performance with untrained input resolution—a significant limitation given the varying resolutions from different scan configurations in our research.

To address this, we adopted the MiT encoder, which offers several advantages. (1) MiT's hierarchical architecture and overlapping patch merging enable the extraction of multi-level feature maps at different resolutions, similar to CNNs. This allows the encoder to capture information at varying scales by combining coarse and fine-grained details. Integrating features from different layers, each capturing specific aspects of the diffraction images, results in rich feature representations essential for accurate image generation. (2) MiT employs Spatial Reduction Attention (Wang et al., 2021), which reduces the spatial dimensions of key and value sequences before the attention operation by a predefined reduction ratio. This approach has been shown to reduce computational and memory costs substantially than traditional self-attention, alleviating the computational bottleneck. (3) The Mix Feed Forward Network (Mix-FFN) replaces fixed PE with a zero-padded $3 \times 3$ convolution. As demonstrated by Islam et al. (2020), zero-padded convolutions implicitly encode positional information. Xie et al. (2021) empirically established that this method not only outperforms fixed PE but is also less sensitive to untrained input resolutions, aiding the handling of multi-resolution inputs in our experiments.

While SegFormer utilizes a multilayer perceptron (MLP) decoder designed for segmentation tasks, this is not optimal for image generation. MLPs inherently lose local spatial coherence due to their fully connected layers, which is detrimental when maintaining spatial relationships are crucial. On the contrary, convolutions preserves locality and are translationally equivariant due to weight sharing and local connectivity. A convolutional decoder can also leverage the multi-level feature maps from the MiT encoder more appropriately than an MLP, effectively integrating spatially localized features. Therefore, we replaced the MLP decoder with a convolutional decoder.

**Feathering.** Our phase retrieval approach reconstructs local patches of the transmission, requiring an effective stitching algorithm. Typically, the patches are placed in their respective coordinates and

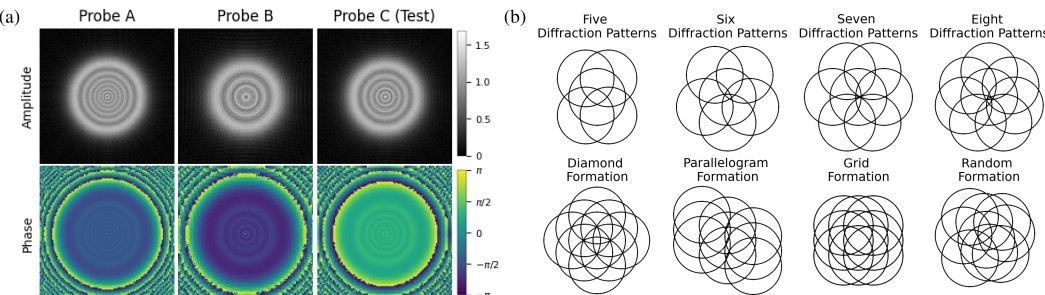

Figure 5: A subset of probe functions we used are shown in (a). Probe A is the primary probe PtychoFormer is trained on and probe B is presented in the finetuning dataset, while probe C is reserved for testing. The grid formation in (b) is used for pre-training, and the other scan configurations are used for finetuning.

average the pixel values of the overlapping regions (Cherukara et al., 2020; Guan & Tsai, 2019; Pan et al., 2023). However, even a slight difference in contrast between adjacent inferences leads to grid artifacts. To avoid this, we use a technique called feathering to ensure smooth transitions between adjacent patches. As visualized in Figure 3(b), we crop the patches to disregard the edges of each patch, where predictions are intentionally omitted, so only the central, well-defined regions of each patch contribute to the final reconstruction. We then linearly taper the pixel values from where the overlap starts up to the edge of the cropped patches. The feathered patches are placed in their respective coordinates, and the pixel values are summed in the overlapping regions, effectively preventing any stitching artifacts (Figure 3(c)).

**Extended-PtychoFormer.** The extended-PtychoFormer (ePF) is our hybrid approach that integrates PtychoFormer and ePIE. One of the challenges in algorithms like ePIE is their susceptibility to convergence on local minima, leading to inaccurate estimates of the transmission function (Maiden et al., 2017). ePF leverages PtychoFormer to provide a well-informed initial estimate through the stitched prediction, serving as a robust starting point for ePIE. This strategy, similar to object initialization (Wittwer et al., 2022a), aims to improve the convergence speed and the reconstruction accuracy.

## 5 EXPERIMENTS

In this section, we present the experimental setup, which includes the dataset creation, training implementation, and evaluation metrics. We then present key advantages of PtychoFormer, such as its tolerance to sparse scan patterns and strong transfer learning capabilities across different probe functions and scan patterns. Following this, we compare PtychoFormer against DL methods like PtychoNN and PtychoNet. Lastly, we discuss how PtychoFormer and ePF compares against the ePIE algorithm.

### 5.1 EXPERIMENTAL SETUP

Lack of training data can be a significant challenge in real-world settings; therefore, we explore the generalization capabilities of PtychoFormer. We devise a pre-training, fine-tuning, and testing pipeline to simulate real-world settings where certain experiments may only have a few samples to fine-tune. We generate the synthetic data and split up the dataset as follows.

**Diffraction Pattern Generation.** The synthetic benchmarks require us to generate diffraction patterns to train and test the model using standard images. Each diffraction pattern requires a pair of images, one for the amplitude and the other for the phase. Each pair of images is used to obtain the transmission function $T(x, y)$ following Eq. 1. Amplitude and phase pixel values are normalized to [0.05, 1] and $[-\pi, \pi]$, respectively.

Each $128 \times 128$ diffraction pattern is generated using Eq. 2 by illuminating $T(x, y)$ with a probe along a prescribed scan configuration, visualized in Figure 5. Since the probe inherently imposes a

finite support constraint, where $T(x, y)$ outside the illumination area is zero, we masked out the corresponding labels, $A(x, y)$ and $\phi(x, y)$, in the regions that are not illuminated. Refer to Appendix B for details on label masking.

**Datasets.** Our training dataset is based on the Flickr30k (Young et al., 2014) dataset. We select 28,600 images from the dataset to create the pre-training and finetuning datasets. Each image is center-cropped, resized, and converted to grayscale during pre-processing. Images are then randomly flipped and rotated to expand the dataset, ultimately generating 62,000 unique pairs of images. The pre-training dataset only uses probe A, presented in Figure 5(a), in a grid scan with varying lateral offsets between diffraction patterns. Specifically, this dataset contains 244,800 samples of 36 different pixel-wise lateral offsets between adjacent scan points, ranging from 20 to 55 pixels (68.7% to 20.3% spatial overlaps).

A key requirement for the model is to adjust to new ptychographic setup conditions without needing a large amount of data. The fine-tuning dataset is designed to introduce new probe functions and scan patterns beyond the training data to adjust the model to new settings. Therefore, we generate fine-tuning samples by introducing 9 new probe functions, including probe B in 5(a), and 7 alternate scan patterns, visualized in Figure 5(b), that are not present in the pre-training dataset.

We devise multiple test sets to perform an ablation study on various scenarios. We hold out 3,100 images from the Flickr30K dataset for testing. We also introduce an unseen probe function (probe C) and a 60-pixel lateral offset grid scan (14.9% spatial overlap) to measure the out-of-distribution performance of our model on unseen setups. To test different domains of inputs, we further evaluate the model on two new datasets, Flower102 (Nilsback & Zisserman, 2008) and Caltech101 (Li et al., 2022). Further details of each dataset can be found in Appendix E.

**Training Details.** We trained PtychoFormer using the mean absolute error (MAE) loss function to minimize the absolute pixel-wise difference between the target label, $Y$, and prediction, $\hat{Y}$, with a batch size of 32. We normalized the phase of $Y$ and $\hat{Y}$ to [0,1] before computing the loss to ensure a balanced loss from the amplitude and phase channel for stable training. See appendix C for improved training curves resulting from phase normalization.

**Evaluation Metrics.** We used MAE and normalized root mean squared error (NRMSE) to separately evaluate the amplitude and phase reconstructions, where $\hat{Y}$ and $Y$ represent the target and estimated values for either amplitude or phase. Similar to Gan et al. (2024), we customized the NRMSE to measure the reconstruction quality after removing the global phase shift. This ensures that our evaluation reflects the true performance of the reconstruction without being skewed by global shifts. Our NRMSE is defined as

$$NRMSE = \frac{\|(\hat{Y} - \tilde{a}) - Y\|}{\|Y\|}, \tag{4}$$

where $\tilde{a}$ corrects the constant phase offset $a$ when computing the NRMSE of phase reconstructions. We determine $\tilde{a}$ as the average pixel-wise difference between $\hat{Y}$ and $Y$. Since the diffraction images in our simulations are centered, we disregard the linear phase term as per Eq. 3 and correct only the constant phase offset using $\tilde{a}$. This term is zero when computing the NRMSE of the amplitude reconstructions.

## 5.2 RESULTS

We investigate the performance of our model using PtychoNN and PtychoNet as our baseline models for our experiments. We also examine the generalization capabilities of PtychoFormer on unseen probe functions and new datasets.

**Comparison with Deep Learning Methods.** We compare PtychoFormer with PtychoNN and PtychoNet. The implementation and training environments of PtychoNN and PtychoNet follow what is outlined in Cherukara et al. (2020) and Guan & Tsai (2019), respectively. PtychoNN and PtychoNet are trained to convergence on the pre-training set and all three models are tested on Flickr30K test set. We can see in Figure 6 that PtychoFormer significantly outperforms both PtychoNN and PtychoNet for both amplitude and phase retrieval. PtychoFormer reduces NMRSE by 25.93% and 41.18% for amplitude reconstruction over PtychoNN and PtychoNet, respectively. All models found phase reconstruction to be more challenging than amplitude, as reflected in the higher

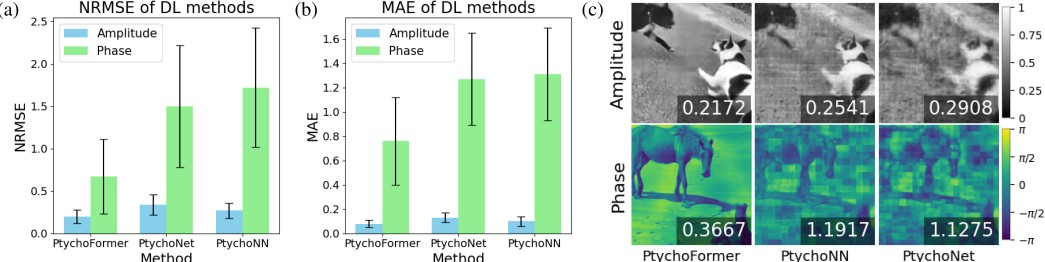

Figure 6: (a) and (b) compare the reconstruction quality of PtychoFormer, PtychoNet, and PtychoNN, measured using normalized root mean squared error (NRMSE) and mean absolute error (MAE), respectively. Results are averaged from 3100 test samples, comprising of 6 × 6 diffraction patterns, each with 20-pixel lateral offsets. (c) depicts the amplitude and phase reconstructions along with the NRMSE values. PtychoFormer consistently outperforms PtychoNN and PtychoNet across all metrics and eliminates the impact of grid artifacts that heavily affect the latter methods.

MAE and NRMSE values for phase. But PtychoFormer performs better phase reconstruction as well as compared to PtychoNN and PtychoNet, reducing NMRSE by 61.05% and 55.33%, respectively. Figure 6(c) also highlights the benefit of feathering, as PtychoNN and PtychoNet are further affected by grid artifacts from their stitching algorithm. PtychoFormer, on the other hand, has a smooth reconstruction along the local prediction boundaries.

**Low to No Data Experiments.** As mentioned before, we require the model to be usable under data-constrained scenarios. We evaluate PtychoFormer under low data availability constraints, where only fine-tuning is possible, as well as no data scenario, where only the pre-trained model is used. For low-data tests, as mentioned before, we introduce new scan patterns and probe functions. For each new configuration, the model is only trained on 2,000 new samples. Remarkably, we see little to no degradation in performance when introducing new scan patterns. The full tables of results are presented in Table 2 and 3 of Appendix F.

We also evaluate PtychoFormer's performance without any further retraining on new data. While our model is only trained on the Flickr30K dataset, we test the model on the Flower102 and Caltech101 datasets. We use probe A and grid scan patterns with 20-pixel lateral offset for the experiments. PtychoFormer achieves $0.18 \pm 0.06$ (amplitude) and $0.51 \pm 0.26$ (phase), and $0.28 \pm 0.09$ (amplitude) and $0.97 \pm 0.65$ (phase) on the Flower102 and Caltech101 datasets, respectively. These results highlight the model's generalized understanding of phase retrieval, even when applied to untrained datasets.

## 5.3 IMPROVEMENTS OVER EPIE

While ePIE is the gold standard for ptychographic phase retrieval, it does have significant drawbacks. The PtychoFormer and ePF are designed to alleviate such drawbacks while achieving comparable reconstruction performance. We compare the performance of PtychoFormer and ePF against ePIE. The ePIE algorithm follows the implementation from Maiden & Rodenburg (2009) and iterates until its objective function, sum squared error (SSE), converges. See Appendix D for implementation details of ePIE.

As mentioned before, a significant drawback of ePIE and similar iterative phase retrieval algorithms is the global phase shift phenomenon. As PtychoFormer is trained in absolute phase values during training, PtychoFormer mitigates global phase shifts. As illustrated in figure 7(f) and (g), ePIE has a high MAE while a comparably lower NRMSE. As the NRMSE is corrected with a global phase shift factor, this shows that ePIE has converged on a shifted phase value. In contrast, PtychoFormer does not suffer from a global phase shift as the MAE and NRMSE are similar. By iteratively improving the PtychoFormer's single-shot prediction, ePF surpasses ePIE on both metrics. Figure 1 visualizes the significant reduction in global phase shifts when using ePF compared to ePIE alone, as seen in both the reconstructed phase images and the corresponding line profiles. However, ePF does not eliminate the global phase shift, as evident in the slight difference between the line profiles of the

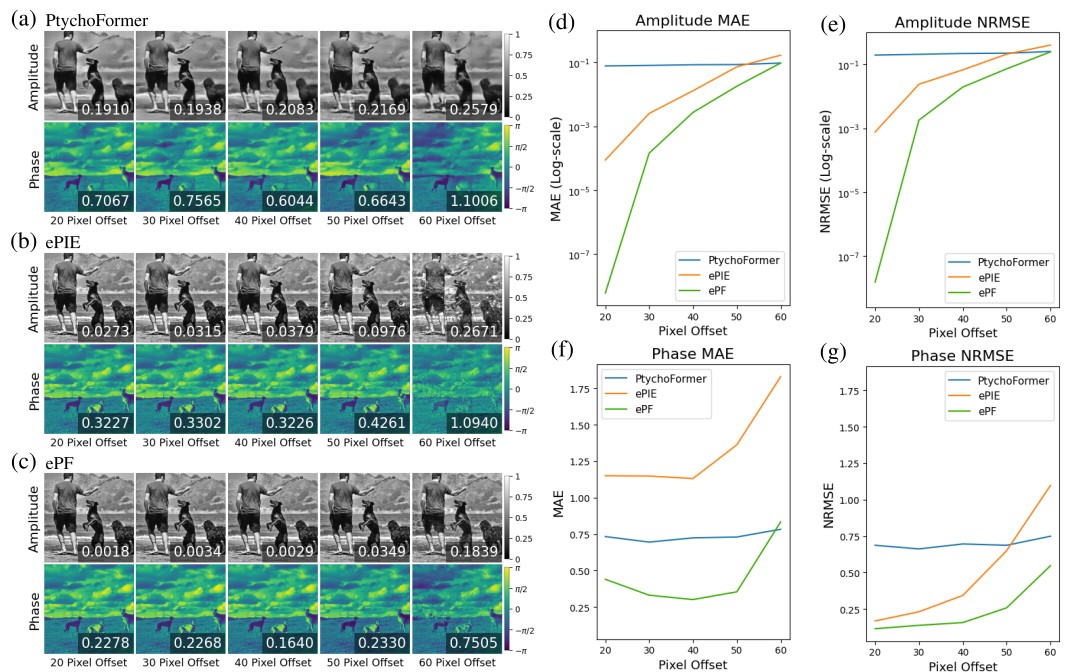

Figure 7: Reconstruction results of PtychoFormer, ePIE, and ePF across various lateral offsets between diffraction patterns are shown in (a), (b), and (c) with normalized root mean squared error (NRMSE) for each reconstruction. (d) and (e) plot the average mean absolute error (MAE) and NRMSE of the amplitude reconstructions and (f) and (g) plot the average MAE and NRMSE of phase reconstructions for each algorithm across different pixel-wise lateral offsets. ePF consistently outperforms ePIE in all metrics and PtychoFormer showcases a robust tolerance to sparser scans than ePIE.

ground truth and ePF. We predict it can be improved with further improvements to the DL-based initialization.

Another critical drawback of ePIE is the computation cost to recover transmission functions. In our experiments with 18×18 diffraction patterns, ePIE, with GPU support, took approximately 0.34 seconds per iteration and 800 to 1500 iterations to converge, resulting in 5 to 8.5 minutes to recover the sample's transmission. In contrast, PtychoFormer completed a one-shot prediction and stitching process in just 0.14 seconds. This translates to a speed-up of 2100 to 3600 times faster than ePIE. Furthermore, ePF reduces the iteration count required for ePIE to converge by approximately 100 iterations in our experiment, leveraging the well-informed estimate from PtychoFormer.

Finally, in Figure 7(a), we observe improved performance over ePIE on sparse scan patterns for both PtychoFormer and ePF. Increasing the offset size reduces the spatial overlap and enables the scanning of a larger area with fewer diffraction patterns. This translates to fewer imaging samples being required and speeding up data collection. The recommended overlap for ePIE is 60% to 70% spatial overlap (Maiden & Rodenburg, 2009) with limited performance beyond. Our methods preserved structural integrity in both the amplitude and phase reconstruction even with a 60-pixel offset (14.9% spatial overlaps) between adjacent patterns. In contrast, ePIE, depicted in Figure 7(b), exhibited significant artifacts at sparser scans. With 60-pixel offset, PtychoFormer outperforms ePIE, as evidenced by its lower MAE and NRMSE values in Figure 7(d–g). All results from various step sizes for ePIE, PtychoFormer, and ePF can be found in Appendix F Table 6.

# 6 DISCUSSION

We can see that PtychoFormer and ePF improve performance in both DL and iterative algorithms. By processing multiple diffraction patterns simultaneously, PtychoFormer captures their positional

relationships, leading to higher-quality reconstructions than models that rely on single diffraction patterns. This spatial awareness stems from our input scheme, which preserves the spatial information of diffraction patterns, and the use of the MiT encoder to effectively capture the spatial relationships well. Our model also generalizes well across different probe functions and scan patterns with minimal fine-tuning and maintains its fidelity even with reduced spatial overlap. ePF uses PtychoFormer, enabling iterative refinement on the data-driven predictions. Importantly, ePF not only outperforms ePIE in terms of reconstruction accuracy but also significantly minimizes global phase shifts, a persistent challenge in phase retrieval.

While we have strong performance on simulated data, transitioning from simulations to real-world applications poses challenges, particularly due to the potential for global phase shifts in the training data. The difficulty lies in the need to train the model on ground truth data generated by conventional algorithms, which may themselves introduce global phase shifts. To address this, calibrated or corrected phase measurements should be used to fine-tune PtychoFormer, allowing it to produce rapid and accurate estimates without global phase shifts. This correction of phase shifts can involve using objects with known phase shifts (Godden et al., 2016), employing interferometric techniques (Cai et al., 2004), or other phase calibration techniques to ensure reliable training data for real-world applications.

## 7 CONCLUSION

PtychoFormer offers a new paradigm in DL phase retrieval for ptychography. It provides a robust and versatile solution capable of processing multiple diffraction patterns while preserving the knowledge of their relative scan points. PtychoFormer enables real-time imaging, which is crucial for applications requiring rapid imaging, but this speed comes at the expense of quality. The model also generalizes well across different probe functions and scan patterns with minimal fine-tuning and maintains its fidelity even with reduced spatial overlap. This adaptability suggests that PtychoFormer could be extended to other imaging modalities requiring phase retrieval, such as X-ray ptychography used to study molecular structures. Moreover, the integration with iterative methods, as seen in ePF, underscores the potential of hybrid approaches in balancing speed and reconstruction quality. ePF allows researchers to obtain a rough estimate of the transmission function in real-time using PtychoFormer, followed by a more accurate refinement using ePIE. While this research has made many significant strides in developing an improved DL phase retrieval algorithm for ptychography, many potential avenues exist to explore. Future work could involve validating PtychoFormer's performance with real ptychographic data and further refining the model to eliminate the need for post hoc corrections.

## REPRODUCIBILITY STATEMENT

We plan to publicly release the code for models, datasets, and experiments represented in the paper.

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

## A  Implementation Details of PtychoFormer

The PtychoFormer architecture leverages a Mix Transformer (MiT) encoder with a convolutional decoder to achieve efficient phase retrieval. Utilizing the MiT-B0 encoder, from the SegFormer architecture (Xie et al., 2021), ensures both computational efficiency and high performance. Key features such as the spatial reduction attention and elimination of fixed positional embedding within the MiT contribute to the efficiency. PtychoFormer processes an input $X \in \mathbb{R}^{9 \times H \times W}$, and outputs $\hat{Y} \in \mathbb{R}^{2 \times H \times W}$, where $H$ and $W$ are the height and width of the input image.

The encoder consists of four stages, each processing the input at different spatial resolutions. These stages incorporate multiple layers of MiT encoder blocks, with 4, 6, 8, and 12 self-attention heads, respectively. In the first stage, the input is processed to $\frac{H}{2} \times \frac{W}{2}$ with 64 feature channels, by performing patch embedding followed by MiT Block 1. The successive MiT stages progressively decrease the resolution by a factor of 2 while increasing the feature channels to 128, 256, and 512, respectively.

The decoder upsamples the outputs from each stage of the encoder using bi-cubic interpolation to $\frac{H}{2} \times \frac{W}{2}$ and passes them through the convolution layer to augment the feature channels to 256. These outputs are concatenated to form a $1024 \times \frac{H}{2} \times \frac{W}{2}$ matrix. Then a convolutional block, followed by a transposed convolutional block, reduces the feature channels and upsamples the resolution to $128 \times H \times W$. Each block is followed by batch normalization and a GELU activation function. The final convolutional layer reduces the feature channels to a $2 \times H \times W$ matrix. The tanh function is applied to the first channel, then multiplied by $\pi$, resulting in a range of $[-\pi, \pi]$. The second channel uses the sigmoid function to yield a range of $[0,1]$. The two output channels correspond to the phase and amplitude estimates, respectively.

## B  Label Detail

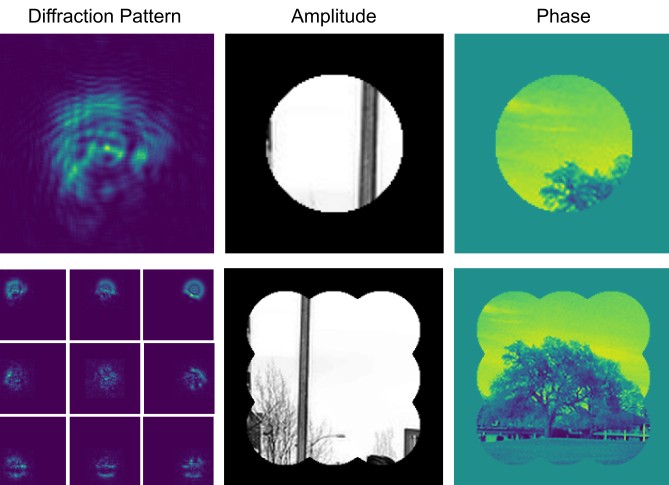

Figure 8: The top row shows the label associated with a single diffraction pattern, where the amplitude and phase are masked to match the illuminated region. The second row displays a 3x3 grid of diffraction patterns in nine separate channels to preserve the spatial dependencies, and the corresponding labels are masked to match the illuminated region as well.

To ensure the model accurately predicts only the information encoded within the diffraction patterns, proper label creation is crucial. Specifically, we mask out the regions in the labels that correspond to areas outside the effective scan point and the radius of the illumination area, preventing the model from learning parts of the transmission function not represented in the diffraction data.

In the top row of Figure 8, you can see an example of a label created for a single diffraction pattern. The label reflects the amplitude and phase values only within the illuminated region, with

areas outside of the illumination masked out to exclude irrelevant information. The second row of Figure 8 demonstrates a 3x3 grid of diffraction patterns, each associated with its respective label. These diffraction patterns are separated into channels, with zero-padding applied to areas outside the diffraction data to maintain consistent input dimensions. This setup enforces the proper learning of the transmission function within the bounds of what the diffraction data encodes.

## C PHASE NORMALIZATION DURING TRAINING

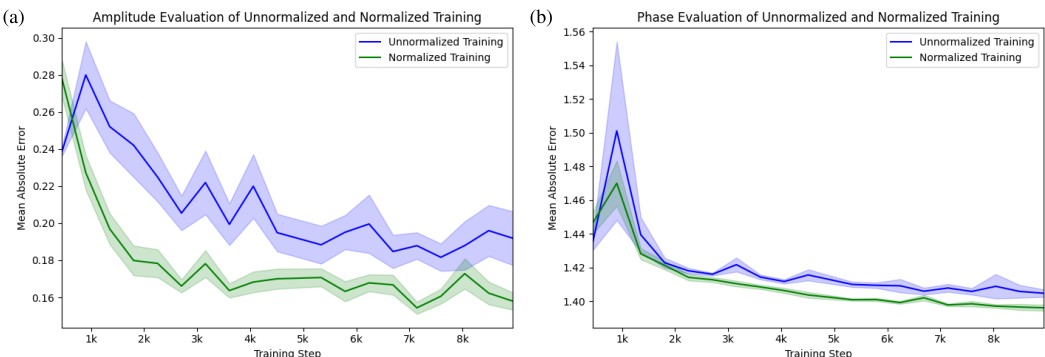

Figure 9: Comparison of average mean absolute error of the test set across training steps for normalized and unnormalized phases during training, based on 20 trials each. The shaded areas represent the standard error. Normalized training exhibits improved generalization and stability than unnormalized training.

During training, we employed a phase normalization technique that scales the phase predictions and labels from their original range of [-$\pi$, $\pi$] to [0,1]. This normalization ensures that the loss values for both amplitude and phase components are within the same range, resulting in similar gradient magnitudes during optimization. Without normalization, computing the mean absolute error (MAE) between images with ranges of [0,1] (amplitude) and [-$\pi$, $\pi$] (phase) would lead to disproportionately larger errors from the phase component due to its wider range. This imbalance can cause the optimization process to focus excessively on minimizing the phase loss, potentially leading to instability during training.

To assess the effectiveness of our normalization approach, we conducted 20 training trials with both unnormalized and normalized phases. Despite normalizing the phase during training, we evaluated the models on the test set using the original unnormalized phase values to measure true performance. We measured the raw MAE scores between the amplitude and phase predictions and their ground truths. The results, shown in Figure 9(a) and (b), indicate that models trained with phase normalization consistently achieve lower MAE on the test set. This demonstrates that phase normalization not only stabilizes the training process but also leads to improved generalization and reconstruction accuracy.

## D EPIE IMPLEMENTATION

The ePIE algorithm refines the estimate of the transmission function, $\hat{T}(x, y)$, through an iterative process. This process aims to minimize the sum of the squared differences between the diffraction patterns produced from the estimated transmission function, $\hat{I}_j(u, v)$, and the measured diffraction pattern, $I_j(u, v)$, at each scanning position $j$.

We suppose the estimated probe measurement, $P(x, y)$, is known and the diffraction s are captured before running ePIE. In our experiment, we set the initial amplitude and phase values of $\hat{T}(x, y)$ to be ones and zeros, respectively.

1. Compute the estimated exit wave: At iteration $k$ for the $j$th scan position, the exit wave is computed as

$$\hat{\psi}_{k,j}(x,y) = \hat{T}_k(x,y) \times P_k(x - X_j, y - Y_j), \quad (5)$$

where $(X_j, Y_j)$ encodes the scanning shifts for the $j$th diffraction pattern. The finite support constraint at each position, $S_j$, is represented as

$$\hat{T}_k(x,y) = \begin{cases} \hat{T}_k(x,y) & \text{if } (x,y) \in S_j \\ 0 & \text{if } (x,y) \notin S_j \end{cases}. \quad (6)$$

2. Update the exit wave: Replace the amplitude of the estimated exit wave with the measured amplitude. First, apply a Fourier transform $F$ to the exit wave:

$$\hat{\Psi}_{k,j}(u,v) = F\left(\hat{\psi}_{k,j}(x,y)\right). \quad (7)$$

Update the exit wave using the measured amplitude:

$$\Psi_{k,j}^{\text{upd}}(u,v) = \frac{\hat{\Psi}_{k,j}(u,v)}{|\hat{\Psi}_{k,j}(u,v)|} \times \sqrt{I_j(u,v)}. \quad (8)$$

Finally, revert to the spatial domain using the inverse Fourier transform:

$$\psi_{k,j}^{\text{upd}}(x,y) = F^{-1}(\Psi_{k,j}^{\text{upd}}(u,v)). \quad (9)$$

3. Update the transmission function estimate using the following equation,

$$\hat{T}_k^{\text{upd}}(x,y) = \hat{T}_k(x,y) + \alpha \frac{\bar{P}_k(x - X_j, y - Y_j)}{|P_k(x - X_j, y - Y_j)|_{\max}^2} \left(\psi_{k,j}^{\text{upd}}(x,y) - \hat{\psi}_j(x,y)\right), \quad (10)$$

where $\bar{P}(x,y)$ is the conjugate of $P(x,y)$ and $\alpha$ is the step size that controls how fast the transmission function is updated. This equation adjusts $\hat{T}_k^{\text{upd}}(x,y)$ based on the difference between the updated exit wave $\psi_{k,j}^{\text{upd}}(x,y)$ and the previous estimate $\hat{\psi}_j(x,y)$. The expression $\frac{\bar{P}_k(x - X_j, y - Y_j)}{|P_k(x - X_j, y - Y_j)|_{\max}^2}$ removes the probe function from the exit wave and is used to weight the update, focusing on areas with stronger illumination. Lastly, we update the transmission function in the area covered by the finite support constraint at position j,

$$\hat{T}_{k+1}(x) = \begin{cases} T_k^{\text{upd}}(x) & \text{if } (x,y) \in S_j \\ \hat{T}_k(x) & \text{if } (x,y) \notin S_j \end{cases}. \quad (11)$$

4. Update the probe function by removing the transmission function from the exit wave, similar to the previous step using the following equation,

$$P_{k+1}(x,y) = P_k(x,y) + \beta \frac{\bar{T}_k(x + X_j, y + Y_j)}{|\hat{T}_k(x + X_j, y + Y_j)|_{\max}^2} \left(\psi_{k,j}^{\text{upd}}(x,y) - \hat{\psi}_j(x,y)\right), \quad (12)$$

where $\bar{T}_k(x,y)$ is the conjugate of $\hat{T}_k(x,y)$ and $\beta$ is the step size for updating the probe function.

5. After iterating through every scan point $j$, increment $k$ and repeat steps $1 - 4$.

6. Repeat the iterative process until the sum squared error function (SSE) converges to a sufficiently small value. This cost function at the $k$th iteration is measured as

$$L[\hat{T}_k(x)] = \sum_j \left(I_j(u,v) - \hat{I}_j(u,v)\right)^2, \quad (13)$$

such that

$$\hat{I}_j(u,v) = |\hat{\Psi}_{k,j}(u,v)|^2. \quad (14)$$

After convergence, the phase of $\hat{T}(x,y)$ is confined between $-\pi$ and $\pi$, resulting in phase wrapping where values exceeding this range. To retrieve the continuous phase, we apply a phase unwrapping algorithm, as introduced by Herráez et al. (2002), which is implemented in the scikit-image library. This post-processing step ensures that the reconstructed phase is free of discontinuities caused by wrapping.

# E  DATASET DETAILS

| Dataset | Source | Probe Functions | Scan Configurations | Lateral Offset |
|---------|--------|-----------------|---------------------|----------------|
| Pre-training | Flickr30k | 1 | Grid | 20 - 55 |
| Fine-tuning | Flickr30k | 2, 3, 4, 5, 6, 7, 8, 9, 10, 11 | Grid | 20 |
| | Flickr30k | 1 | 5 - 8 Diffraction Patterns, Diamond, Random, Grid, Parallelogram | – |
| Test | Flickr30k | 1 | Grid | 20, 30, 40, 50, 60 |
| | Flickr30k | 2, 3, 4, 5, 6, 7, 8, 9, 10, 11, Test Probe | Grid | 20 |
| | Flickr30k | 1 | 5 - 8 Diffraction Patterns, Diamond, Random, Parallelogram | – |
| | Flower102 | 1 | Grid | 20 |
| | Caltech101 | 1 | Grid | 20 |

Table 1: Details of the pre-training, finetuning, and test datasets are showcased in this table. For each dataset, we present the source of the raw images and the probe functions, scan configurations, and pixel-wise lateral offsets the dataset contains.

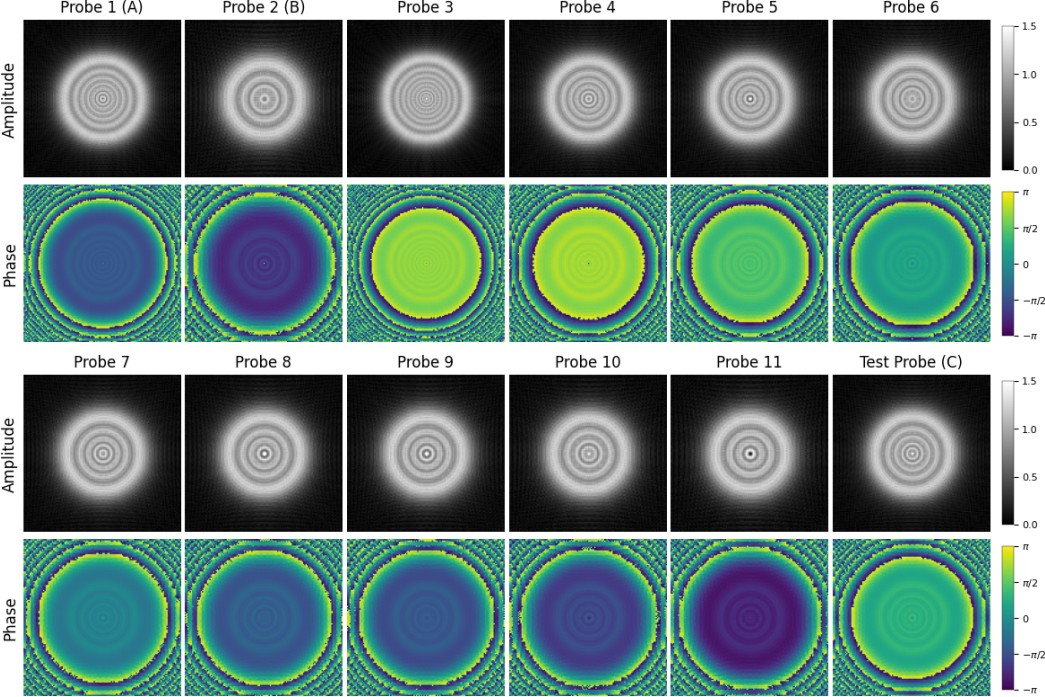

Figure 10: Amplitude and phase profiles for 12 probe functions, including the Test Probe. Probes 1 and 2 are referred to as Probe A and Probe B in the main body of the paper, while the Test Probe is referred to as Probe C. Each probe demonstrates variations in amplitude and phase, which are used to train the model to be resilient to different probe functions.

This appendix and Table 1 provide detailed information about the pre-training, fine-tuning, and test datasets. As a preliminary, probes A, B, and C, from the main body of text, are relabeled as we present all 12 probe functions. As visualized in Figure 10, the 11 training probes are numerically labeled from 1 - 11, such that probes 1 and 2 correspond to probes A and B, and probe C is labeled as the test probe.

The Flickr30k (Young et al., 2014) is the primary dataset we used in our experiment, where we allocated 28,600 images for training and 3,100 images for testing. The pre-training dataset consists of all training images to generate diffraction patterns using a grid scan pattern with 36 different lateral offsets between adjacent scan points, ranging from 20 to 55 pixels. These offsets correspond to spatial overlaps between diffraction patterns ranging from 68.7% to 20.3%. The pre-training dataset consists of these 36 configurations, each containing 6,800 triplets of nine simulated diffraction patterns along with the corresponding amplitude and phase labels, totaling 244,800 data points. We only used probe 1 in this dataset.

The finetuning dataset uses a subset of the training images to create a smaller dataset that introduces 7 additional scan patterns and 9 new probe functions (i.e. probe 2 - 11). The scan patterns included various configurations using probe 1, from 5 to 8 diffraction patterns and grid, diamond, parallelogram, and random layouts using 9 diffraction patterns. Probes 2 - 11 were used in a grid scan with a constant lateral offset of 20 pixels. The finetuning dataset included 2,000 samples per configuration, totaling 36,000 data points.

We created five test datasets, out of which three were created using the allocated test images from Flickr30k, to evaluate the model's performance under various scenarios, including out-of-distribution conditions. The first test dataset consists of grid scan with lateral offsets of 20, 30, 40, 50, and 60 pixels between adjacent scan points. These offsets correspond to spatial overlaps of 68.7%, 53.6%, 39.4%, 26.3%, and 14.9%, respectively. We created 100 samples for each of these grid scans. The second dataset contains grid scans of 20 pixel offset using probes 2 - 11, and the test probe, each with 200 samples. The third dataset includes configurations using the seven new scan patterns introduced in the fine-tuning dataset. To assess the model's ability to generalize to different domains, we further evaluated it on two new datasets: Flower102 (Nilsback & Zisserman, 2008) and Caltech101 (Li et al., 2022). We created 8,000 samples from each dataset.

# F    SIMULATION RESULTS

This section presents the empirical results from our simulations, providing the exact mean and standard deviation values for various probe functions, scan configurations, and comparisons of different phase retrieval methods. Tables 4 and 6 are included here in the appendix, as the corresponding results are visualized as charts in the main body of the paper.

We evaluated 12 distinct probe functions corresponding to different illumination patterns during diffraction, detailed in Table 2. Additionally, Table 3 presents the results from eight different scan configurations, demonstrating how the geometry of diffraction scans influences reconstruction accuracy. Notably, the grid formation yielded the lowest errors for both amplitude and phase reconstructions. This is likely because the model was primarily trained on grid configurations and only later fine-tuned on other scan setups, giving it an advantage when reconstructing from grid data compared to more irregular formations, such as random configurations.

Table 4 compares the performance of PtychoFormer with two other deep learning-based methods, PtychoNet and PtychoNN, under 20-pixel offsets between each diffraction pattern. PtychoFormer consistently demonstrates superior performance in both amplitude and phase reconstructions, achieving lower MAE and NRMSE values. Moreover, this trend continues across larger offsets, even on an unseen lateral offset (i.e. 60 pixels), shown in Table 5. In Table 6, we compare the performance of PtychoFormer, extended-PtychoFormer (ePF), and ePIE across different lateral offsets between diffraction patterns. As discussed in the main body, PtychoFormer exhibits persistent performance across different offsets, while ePIE showcases severe degradation. The degradation is present for ePF, but consistently outperforms ePIE.

Fourier Ring Correlation (FRC) analysis (Koho et al., 2019), shown in Figure 11, reveals that PtychoFormer effectively recovers low to midrange spatial frequencies, but struggles with high-frequency details. The 1/7 threshold is an empirically validated criterion for distinguishing between

| Probe Function | Amplitude NRMSE | Phase NRMSE |
|---|---|---|
| Probe 1 (A) | 0.24 ± 0.10 | 0.79 ± 0.52 |
| Probe 2 (B) | 0.28 ± 0.10 | 0.95 ± 0.74 |
| Probe 3 | 0.23 ± 0.11 | 0.81 ± 0.58 |
| Probe 4 | 0.25 ± 0.11 | 0.76 ± 0.61 |
| Probe 5 | 0.24 ± 0.11 | 0.76 ± 0.74 |
| Probe 6 | 0.25 ± 0.10 | 0.87 ± 0.81 |
| Probe 7 | 0.25 ± 0.11 | 0.83 ± 0.68 |
| Probe 8 | 0.27 ± 0.09 | 0.83 ± 0.62 |
| Probe 9 | 0.27 ± 0.10 | 1.00 ± 0.82 |
| Probe 10 | 0.29 ± 0.11 | 1.11 ± 1.22 |
| Probe 11 | 0.28 ± 0.11 | 1.02 ± 0.84 |
| Test Probe (C) | 0.22 ± 0.10 | 0.83 ± 0.77 |

Table 2: Normalized root mean squared error (NRMSE) values for 11 probe functions used during training and the test probe used exclusively for testing. The table presents the NRMSE for both amplitude and phase reconstructions, demonstrating the model's ability to generalize across various probe functions, including an unseen probe function.

| Scan Configuration | Amplitude NRMSE | Phase NRMSE |
|---|---|---|
| Five Diffraction Patterns | 0.27 ± 0.11 | 0.71 ± 0.37 |
| Six Diffraction Patterns | 0.27 ± 0.09 | 0.69 ± 0.36 |
| Seven Diffraction Patterns | 0.26 ± 0.10 | 0.68 ± 0.44 |
| Eight Diffraction Patterns | 0.27 ± 0.10 | 0.77 ± 0.46 |
| Diamond Formation | 0.26 ± 0.09 | 0.67 ± 0.46 |
| Parallelogram Formation | 0.29 ± 0.12 | 0.71 ± 0.45 |
| Grid Formation | 0.20 ± 0.09 | 0.66 ± 0.32 |
| Random Formation | 0.26 ± 0.08 | 0.72 ± 0.43 |

Table 3: Amplitude and phase normalized root mean squared error (NRMSE) values for various scan configurations using probe A are presented. Values represent the average NRMSE of 100 test data by comparing the reconstructed central squares to the ground truth. PtychoFormer shows similar performance across different scan configurations even with minimal training.

| Method | Amplitude | | Phase | |
|---|---|---|---|---|
| | MAE | NRMSE | MAE | NRMSE |
| PtychoFormer | **0.08 ± 0.03** | **0.20 ± 0.08** | **0.76 ± 0.36** | **0.67 ± 0.44** |
| PtychoNet | 0.13 ± 0.04 | 0.34 ± 0.12 | 1.27 ± 0.38 | 1.50 ± 0.72 |
| PtychoNN | 0.10 ± 0.04 | 0.27 ± 0.09 | 1.31 ± 0.38 | 1.72 ± 0.7 |

Table 4: Mean absolute error (MAE) and normalized root mean squared error (NRMSE) for amplitude and phase reconstructions of three deep learning models showcases PtychoFormer's exceptional performance. Results are averaged from 3,100 test images, each comprising $6 \times 6$ diffraction patterns with 20-pixel lateral offsets.

| Method | Amplitude | | Phase | |
|---|---|---|---|---|
| | MAE | NRMSE | MAE | NRMSE |
| PtychoFormer | **0.10 ± 0.03** | **0.26 ± 0.06** | **0.82 ± 0.23** | **0.76 ± 0.33** |
| PtychoNet | 0.17 ± 0.03 | 0.53 ± 0.07 | 1.35 ± 0.21 | 1.45 ± 0.32 |
| PtychoNN | 0.15 ± 0.03 | 0.47 ± 0.05 | 1.35 ± 0.21 | 1.71 ± 0.39 |

Table 5: Mean absolute error (MAE) and normalized root mean squared error (NRMSE) for amplitude and phase reconstructions are taken by averaging the performance across 3,100 test images, each comprising $6 \times 6$ diffraction patterns with 60-pixel lateral offsets. Even on unseen scan, PtychoFormer outperforms PtychoNN and PtychoNet.

| Method | Lateral Offset | Amplitude | | Phase | |
|---|---|---|---|---|---|
| | | MAE | NRMSE | MAE | NRMSE |
| PtychoFormer | 20 | 0.08 ± 0.04 | 0.20 ± 0.09 | 0.75 ± 0.33 | 0.66 ± 0.32 |
| ePF | 20 | **5.89e-9 ± 3.72e-9** | **1.51e-8 ± 2.33e-8** | **0.44 ± 0.64** | **0.12 ± 0.19** |
| ePIE | 20 | 8.74e-5 ± 5.62e-4 | 7.66e-4 ± 4.50e-3 | 1.15 ± 0.93 | 0.17 ± 0.30 |
| PtychoFormer | 30 | 0.08 ± 0.03 | 0.21 ± 0.08 | 0.70 ± 0.29 | 0.66 ± 0.41 |
| ePF | 30 | **1.44e-4 ± 4.22e-4** | **1.81e-3 ± 5.23e-3** | **0.33 ± 0.46** | **0.14 ± 0.20** |
| ePIE | 30 | 2.52e-3 ± 3.46e-3 | 0.02 ± 0.03 | 1.15 ± 0.90 | 0.23 ± 0.32 |
| PtychoFormer | 40 | 0.09 ± 0.03 | 0.22 ± 0.07 | 0.72 ± 0.26 | 0.70 ± 0.36 |
| ePF | 40 | **2.72e-3 ± 2.77e-3** | **0.02 ± 0.02** | **0.30 ± 0.41** | **0.16 ± 0.19** |
| ePIE | 40 | 0.01 ± 0.01 | 0.07 ± 0.04 | 1.13 ± 0.99 | 0.34 ± 0.35 |
| PtychoFormer | 50 | 0.09 ± 0.03 | 0.23 ± 0.07 | 0.73 ± 0.25 | 0.69 ± 0.30 |
| ePF | 50 | **0.02 ± 0.01** | **0.07 ± 0.04** | **0.35 ± 0.68** | **0.26 ± 0.23** |
| ePIE | 50 | 0.07 ± 0.04 | 0.21 ± 0.10 | 1.36 ± 1.34 | 0.65 ± 0.32 |
| PtychoFormer | 60 | 0.10 ± 0.03 | 0.26 ± 0.07 | **0.78 ± 0.21** | 0.75 ± 0.31 |
| ePF | 60 | **0.09 ± 0.04** | **0.24 ± 0.10** | 0.83 ± 0.96 | **0.55 ± 0.26** |
| ePIE | 60 | 0.17 ± 0.04 | 0.41 ± 0.08 | 1.83 ± 1.02 | 1.10 ± 0.21 |

Table 6: Comparison of reconstruction performance for PtychoFormer, ePF, and ePIE across different lateral offsets highlights the remark performance of ePF. The table reports mean absolute error (MAE) and normalized root mean squared error (NRMSE) for amplitude and phase reconstructions. Results are averaged from 100 test datasets, each consisting of 6 × 6 diffraction patterns.

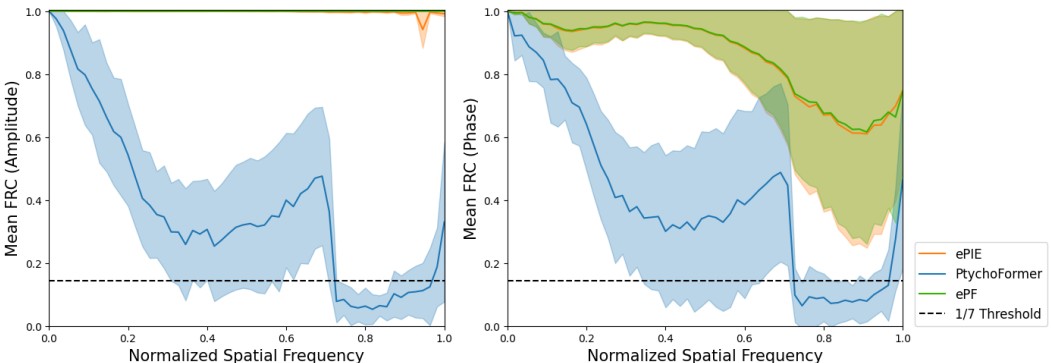

Figure 11: Average Fourier Ring Correlation (FRC) of 100 test samples under 20-pixel lateral offsets to illustrate the correlation between the reconstruction and the ground truth across different spatial frequencies. Shaded areas of the graph represent the standard deviation of FRC values. PtychoFormer exhibits a high correlation in the low and mid spatial frequencies while showing a dip in high spatial frequency. However, ePF showcases a near perfect amplitude correlation and a subtle improvement in phase correlation than ePIE.

meaningful signal and noise. The average FRC curves for both amplitude and phase drop below the threshold at normalized spatial frequencies of 0.71 to 0.95, highlighting its limitations in capturing fine details. However, ePF showcases better FRC values than ePIE. Notably, ePF exhibits near-perfect mean FRC of the amplitude reconstruction and slight improvement in the phase.

