# OpenReview forum: "PtychoFormer: A Transformer-based Model for Ptychographic Phase Retrieval"
_ICLR.cc/2025/Conference — Submitted to ICLR 2025_

### Official Review · Reviewer_1WZv · 2024-10-28

**Soundness:** 2
**Presentation:** 1
**Contribution:** 2
**Rating:** 3
**Confidence:** 4

**Summary:**

The authors propose a transformer-based method for ptychographic phase retrieval called PtychoFormer, and they propose to use it as an initialization for the existing ePIE method.

**Strengths:**

The proposed PtychoFormer runs quickly and yields performance that is 1 or 2 dB better than two deep-network approaches from about 5 years ago.

**Weaknesses:**

The problem statement and model assumptions are extremely vague.
* The background section never actually states which quantities are known, which are to be estimated, and which are nuisance parameters.  The experiment section seems to suggest that the goals of estimation are the amplitude and phase of the “transmission function,” but that leaves major questions as to whether the complex probe function and the lateral offsets Xj and Yj are perfectly known or not.
* The authors also never describe the prior knowledge on the amplitude and phase images. Figure 2 suggests that they are both natural grayscale images, yet completely unrelated, and it seems that the same convention was followed in the experiments. But it is difficult to understand how and why this would manifest in any practical setting.
* The authors never describe why there is no measurement noise.
* The representation of all quantities as continuous functions of location (x,y) is confusing and unnecessary, since in reality these locations are sampled on a discrete grid.  The authors never actually state this, which further increases the confusion.
* In the end, it seems that measurement vector (across all diffraction patterns) can be written abs(A*t) for known matrix A and unknown complex vector t, which means that it is a standard phase-retrieval problem that can be solved using a huge number of methods, not some specialized phase-retrieval problem.

The discussion of “the current state” of phase-retrieval algorithms is lacking, as are the methods considered for the numerical experiments.
* Well-known classical algorithms like HIO from the 80s are missing.
* The family of plug-and-play algorithms like prDeep, Deep-ITA, etc., are never mentioned.
* Recent deep-learning based approaches are not mentioned, such as those based on diffusion.
* Because all of the aforementioned methods jointly would process all diffraction patterns, they would likely substantially outperform any method than handles each diffraction pattern separately, such as PtychoNet, PtychoNN, and PtyNet.
* Likewise, if PtychoDV does not property model the probe function, which seems to be what the authors are suggesting around line 192, then all of these aforementioned methods would likely substantially outperform PtychoDV as well.

The proposed methods have strong limitations.
* PtychoFormer is heavily dependent on the assumption that the amplitude and phase of the transmission function are two unrelated natural images, and that massive training sets are available for both.  No effort to justify this assumption is given.  In practice, the amplitude and phase images will be highly interrelated, neither will be a natural image, and massive training sets will not be available.
* PtychoFormer is heavily dependent on the absence of measurement noise, which itself is never justified.
* The ePF method is intellectually trivial: just initialize ePIE with PtychoFormer.  One could just as easily initialize ePIE with existing DL methods, which perform only slightly worse than the proposed PtychoFormer according to Figure 6.

The experimental results are problematic.
* The PtychoNet and PtychoNN competitors appear to be very weak, as described earlier. Also, there are only two competitors tested, which is far fewer than in most ICLR papers.  The PtychoDV method, which is much more closely related to the proposed PtychoFormer (since it is based on ViT) is not investigated.
* According to Figure 6, the proposed PtychoFormer reduces NRMSE in amplitude recovery by only 1.3dB (i.e., 26%) relative to PtychoNN, which is a primitive method that processes diffraction patterns one at a time.  This does not seem impressive.
* As for phase reconstruction, the competing methods in Figure 6 all produce NRMSEs that are greater than 1.  But how could this be?  A trivial method that always reports zero for the phase image would give an NRMSE of one.  As a result, this casts doubt the implementations of the competing methods.  Furthermore,  it means that PtychoFormer only provides a small benefit (25% reduction?) over the trivial all-zeros method, which again does not seem impressive.
* Moving on to Figure 7, the claim in subplots (d) and (e) that the proposed ePF method achieves an average 1e-8 recovery MAE and NRMSE is simply impossible.  The impossibility can be verified by looking at the example of NRMSE at 20 pixel offset in subplot (c), which is 5 orders of magnitude larger!

**Questions:**

* Which practical applications cause a natural image to manifest as the amplitude and an unrelated natural image to manifest as the phase?
* Is the light probe P really perfect known as you seem to assume?  If so, why does ePIE iteratively approximate it?

---

### Official Review · Reviewer_PSVU · 2024-11-04

**Soundness:** 2
**Presentation:** 2
**Contribution:** 1
**Rating:** 5
**Confidence:** 4

**Summary:**

The manuscript presents PtychoFormer, a hierarchical transformer-based model aimed at enhancing deep learning phase retrieval for ptychography. While it offers a framework that processes multiple diffraction patterns and maintains spatial awareness through relative scan point information.

**Strengths:**

The model can achieve faster imaging speeds.

The development of the extended-PtychoFormer (ePF) hybridizes deep learning with iterative methods, improves reconstruction quality.

**Weaknesses:**

The novelty of the proposed approach appears to be limited, as it does not present substantial advancements over existing methodologies.

This narrow comparison does not provide a comprehensive assessment of its performance against the broader landscape of existing approaches, raising questions about the validity and significance of the claims.

**Questions:**

While the introduction of PtychoFormer offers some improvements, such as processing multiple diffraction patterns. However, it can be argued that the model largely represents a combination of existing modules rather than a novel approach.

The comparative analysis shown in Figure 6 is inadequate, as the proposed method is evaluated against only two other techniques, which does not provide a comprehensive assessment of its performance. A broader comparison with additional state-of-the-art methods would strengthen the validity of the claims made in this study.

---

### Official Review · Reviewer_La6D · 2024-11-05

**Soundness:** 2
**Presentation:** 3
**Contribution:** 2
**Rating:** 3
**Confidence:** 4

**Summary:**

This paper introduces PtychoFormer for data-driven, single-shot ptychographic phase retrieval. The model is robustness with
sparsely scanned diffraction patterns and achieves imaging speeds up to 3600 times faster than the extended ptychographic iterative
engine (ePIE). Additionally, the authors present the extended-PtychoFormer (ePF), a hybrid model that merges the strengths of
PtychoFormer and ePIE, effectively minimizing global phase shifts and significantly improving reconstruction quality.

**Strengths:**

1. The paper is written in a logical sense and is easy to follow.
2. The author introduces PtychoFormer as a new approach to achieve the fast speed of phase retrieval.
3. The authors provide simulated results to demonstrate the performance and speed of their proposed algorithm.

**Weaknesses:**

The author's contributions and innovations in the PtychoFormer appear trivial and insufficient, the network structure is similar to previous Mix Transformer (MiT). To enhance the impact of this work, I recommend that the author clearly articulates the unique contributions and advancements of the PtychoFormer .

**Questions:**

As I mentioned in the weakness part, the innovation presented in the PtychoFormer may not be robust enough to meet the standards typically expected for an ICLR paper. I suggest the author to provide a more detailed clarification of the contributions in the rebuttal. It would be beneficial to highlight specific aspects of the model that distinguish it from existing work.

---

### Official Review · Reviewer_r8CE · 2024-11-06

**Soundness:** 1
**Presentation:** 2
**Contribution:** 1
**Rating:** 1
**Confidence:** 4

**Summary:**

This paper develops a transformer based network for solving the ptychographic phase retrieval problem. The proposed method is tested on simulated data and outperforms two CNN baselines. It is also compared with a conventional algorithm ePIE, which it generally underperforms. However, one can initialize the iterative algorithm with the output of the transformer to get results that are better than either.

**Strengths:**

+++Works better than two NN baseline methods.

**Weaknesses:**

---The global phase ambiguity is an unresolvable problem---without modifying the measurement process, there is no way to know the phase of the light that hits the microscope. A global phase shift would have zero impact on the intensity-only measurements. Networks are just hallucinating one possible solution.

---Any solution that is equivalent up to a global phase shift should be considered equivalent. The results in figure 1 are extremely misleading. The ePIE solution is just as correct as the others.

---Tested only on simulated data. Simulated data (natural images) does not match the statistics of typical ptychography samples

---Missing comparison: PtychoDV (Gan et al.) recently applied vision transformers to solve the ptychography problem. The argument for why this method doesn't capture spatial relationships ("coordinates alone inadequately capture the overlap between the vectorized patterns with high granularity") is unconvincing and there is no comparisons with this method.

---Novelty: If I understand correctly, ePF is just using Ptychoformer output to initialize a conventional iterative algorithm

**Questions:**

What differentiates this method from PtychoDV?

Why is the global phase ambiguity problem a problem? When would you ever care about global phase? How is the proposed method doing more than hallucinating the global phase?

---

### Official Review · Reviewer_1DEg · 2024-11-07

**Soundness:** 2
**Presentation:** 3
**Contribution:** 2
**Rating:** 5
**Confidence:** 3

**Summary:**

This paper studies ptychographic phase retrieval and proposes a transformer-based model for recovering phase and amplitude from a series of diffraction patterns. The method leverages a Mix Transfomer (MiT) with hierarchical architecture to extract features at different resolutions and uses a convolutional decoder to reconstruct local patches of the transmission. The patches are stitched together with feathering and form the final reconstruction. The improved performance over other deep methods is demonstrated on simulated data. The output can be further used as an initialization for ePIE, which outperforms ePIE.

**Strengths:**

The proposed transformer-based method outperforms other DL methods. The performance gain could be a more effective MiT encoder than CNN, the feathering technique when stitching local patches, etc.

**Weaknesses:**

* The performance gain with the proposed method seems marginal. The advantage of using transformer is not made very clear.
* One major concern is whether the size of training data is enough to train a vision transformer. As the author mentioned, lack of training data is a significant challenge in real world, and the pretraining and finetuning datasets are from <100K images. Is data with this size large enough for the pretraining of a vision transformer to outperform CNN? Will adopting some pretrained ViT/MiT and finetuning from there work better?
* Using the output of PtychoFormer as the initialization for ePIE is not very convincing to demonstrate the effectiveness of PtychoFormer. It would be better to benchmark it with using the output of other models (PtychoNet, PtychoNN, etc) as initializations for ePIE.

**Questions:**

* Maybe I missed this part: how does the proposed model eliminate global phase shift by design?
* What's the speed difference between ePIE and ePF?

---

### Meta-Review · Area_Chair_Auxc · 2024-12-20

**Metareview:**

The paper proposes Ptychoformer, a hirerarchical transformer-based method for solving the ptychographic phase retrieval problem.

The algorithm runs quickly and outperforms earlier (but somewhat outdated) neural-network based methods for phase retrieval.

The reviewers raised several (mostly unanimous) concerns. The idea seems somewhat incremental, the baselines are somewhat outdated and weak, and the performance gains over said baselines were marginal. Moreover, the evaluations were only on simulated data on natural images, so the applicability of the resulting model to real-world ptychographic microscopy problems is questionable.

**Additional Comments On Reviewer Discussion:**

There was no response from the authors.

---

### Decision · Program_Chairs · 2025-01-22

Reject